# Investigating the Molecular Impact of GGMSC on Redox and Metabolic Pathways in Pancreatic Cancer Cells

**DOI:** 10.3390/antiox14101163

**Published:** 2025-09-25

**Authors:** Arun Kumar Selvam, Mehran Ghaderi, Joakim Dillner, Shaheen Majeed, Mikael Björnstedt

**Affiliations:** 1Department of Laboratory Medicine, Division of Pathology F46, Karolinska University Hospital Huddinge, SE-141 86 Stockholm, Sweden; 2Department of Clinical Science, Intervention and Technology (CLINTEC), ANA Futura, Karolinska Institutet, SE-141 52 Stockholm, Sweden; mehran.ghaderi@ki.se (M.G.); joakim.dillner@ki.se (J.D.); 3Department of Clinical Pathology and Cancer Diagnostics, Karolinska University Hospital, SE-141 86 Stockholm, Sweden; 4Sabinsa Corporation, 20 Lake Drive, East Windsor, NJ 08520, USA; shaheen@sabinsa.com

**Keywords:** gamma-glutamyl-selenomethylselenocysteine, pancreatic ductal adenocarcinoma, selenium cytotoxicity, redox regulation, ferroptosis, epigenetic modification

## Abstract

Pancreatic ductal adenocarcinoma (PDAC) remains a highly aggressive malignancy with limited treatment options. Targeting metabolic vulnerabilities and disrupting redox stress pathways has gained increasing attention as a potential therapeutic strategy. γ-Glutamyl-selenomethylselenocysteine (GGMSC) is a selenium-containing compound structurally related to seleno-L-methylselenocysteine (MSC), which has shown anticancer potential in preclinical models, although its molecular effects in PDAC are not well defined. In this study, we investigated the transcriptomic response to high-dose GGMSC in two PDAC cell lines, CAPAN-2 and HPAF-II. RNA sequencing and cytotoxicity assays revealed marked sensitivity to GGMSC in CAPAN-2 cells, associated with activation of oxidative stress and ferroptosis-related pathways, alongside downregulation of metabolic and cell cycle genes. Conversely, HPAF-II cells displayed limited transcriptional alterations and maintained proliferative and metabolic programs. These findings offer insights into the molecular mechanisms underlying GGMSC-induced transcriptional responses in PDAC and suggest potential avenues for future investigations of selenium-based therapies in pancreatic cancer.

## 1. Introduction

Pancreatic ductal adenocarcinoma (PDAC) is among the most aggressive and lethal malignancies, characterized by late diagnosis, high metastatic potential, and resistance to conventional therapies. Despite advances in surgical techniques and chemotherapeutic regimens, the five-year survival rate for PDAC remains below 10% [1,2,3]. The dense stromal environment, metabolic plasticity, and redox imbalances within PDAC tumors contribute to their poor therapeutic responsiveness and highlight the need for novel treatment approaches [4]. Targeting metabolic vulnerabilities and oxidative stress has emerged as a promising strategy in PDAC therapy. Recent studies have shown that pathways regulating redox homeostasis, like the KEAP1-NRF2 axis, and cell death mechanisms like ferroptosis, play critical roles in PDAC progression and therapeutic resistance [5]. Agents capable of simultaneously modulating redox homeostasis and inducing ferroptosis may offer therapeutic benefits by selectively impairing tumor cell survival mechanisms while sparing normal cells.

Selenium-based compounds have gained significant interest due to their ability to modulate oxidative stress, inhibit histone deacetylases (HDACs), and trigger multiple forms of cell death, including apoptosis and ferroptosis [6]. Among these, seleno-L-methylselenocysteine (MSC) has demonstrated promising anticancer efficacy in multiple preclinical models by inducing apoptosis, ferroptosis, and ER stress, and by inhibiting HDAC activity [7,8]. A structurally related analog, γ-glutamyl-selenomethylselenocysteine (GGMSC), has also shown anticancer potential [9]. GGMSC is a naturally occurring organoselenium compound predominantly found in selenium-enriched garlic, in which it represents a major selenium species [10]. It has also been identified in selenium-accumulator plants such as *Astragalus bisulcatus* [11]. The natural occurrence of GGMSC and related selenoamino acids supports their biological relevance and highlights their potential in chemopreventive applications.

GGMSC, a γ-glutamylated analog of MSC, is proposed to act as a prodrug, releasing bioactive selenium metabolites upon enzymatic cleavage. Although studies on GGMSC remain limited, available evidence suggests that it shares many mechanistic properties with MSC [10]. In rodent models, GGMSC has demonstrated comparable tumor-suppressive activity and gene expression profiles to MSC, including inhibition of tumor growth and modulation of redox pathways [10,12,13]. To date, the only in vitro study that has tested GGMSC was performed in COLO205 colon adenocarcinoma cells, where concentrations up to 250 µM did not produce significant cytotoxicity [9]. In contrast, MSC in the same study triggered apoptosis through ER stress and death receptor pathways. The in vivo study of GGMSC by Dong et al. [10] further demonstrated that GGMSC is bioavailable from garlic and exhibits elimination kinetics and tumor-suppressive activity comparable to MSC in rodent models. Collectively, these findings suggest that GGMSC is biologically active but may require higher concentrations, prolonged exposure, or specific enzymatic activation contexts to exert cytotoxic effects.

Given these observations, we aimed to investigate the transcriptomic effects of GGMSC in PDAC cells, with a focus on redox signaling, ferroptosis, metabolic remodeling, and cell cycle regulation. We selected CAPAN-2 and HPAF-II cell lines as representative of PDAC models with distinct origins and molecular features. CAPAN-2 is a well-differentiated adenocarcinoma line derived from a primary pancreatic tumor and carries common PDAC driver mutations including KRAS, TP53, and SMAD4. In contrast, HPAF-II is a well-differentiated line derived from metastatic ascites, also harbouring KRAS and TP53 mutations, and is characterized by strong epithelial morphology, with metastatic potential. These contrasting features provided the rationale for comparing GGMSC responses in primary- versus metastasis-derived PDAC contexts. By analyzing the transcriptional responses of CAPAN-2 and HPAF-II to high-dose GGMSC treatment, we sought to elucidate the molecular mechanisms underlying its cytotoxic effects and to identify potential factors contributing to differential sensitivity. This study further provides insight into the therapeutic potential of GGMSC as a redox-modulating agent in PDAC and its relation to MSC-based selenium therapies.

## 2. Materials and Methods

### 2.1. Cell Culture and Growth Conditions

The human pancreatic cancer cell lines HPAF-II and CAPAN-2 were obtained from the American Type Culture Collection (ATCC; CRL-1997, Wesel, Germany) and the German Collection of Microorganisms and Cell Cultures (DSMZ; ACC-245, Braunschweig, Germany), respectively. Cells were maintained in Eagle’s Minimum Essential Medium (EMEM; ATCC) supplemented with 10% heat-inactivated fetal bovine serum (FBS; Gibco, Paisley, UK) and incubated at 37 °C in a humidified atmosphere containing 5% CO_2_. Antibiotics were not included in the culture medium. Mycoplasma contamination was routinely assessed. Cell numbers were determined using a TC20™ automated cell counter (Bio-Rad, Portland, ME, USA).

Prior to RNA sequencing experiments, a stepwise serum re-adaptation protocol was employed to minimize biological variability due to asynchronous cell cycling. Cells were initially cultured in EMEM without fetal bovine serum (0% FBS) to induce a transient quiescent (G0) state. Subsequently, FBS concentration was gradually increased (2%, 5%, and finally 10%), with each condition maintained for 72 h to allow controlled re-entry into the cell cycle. Following complete re-adaptation to 10% FBS, cells were cultured for three additional passages under standard conditions before GGMSC treatment and RNA isolation. Similar serum starvation and refeeding protocols have been used in previous transcriptomic and proteomic studies to synchronize cancer cells and reduce background transcriptional variability [14]. This approach helped synchronize the cellular population and ensured that most cells were in a comparable growth phase at the time of treatment, thereby reducing transcriptional noise and improving the robustness of downstream RNA-seq analysis.

### 2.2. GGMSC Treatment and Cytotoxicity Assay

γ-Glutamyl-selenomethylselenocysteine (GGMSC) was supplied by Sabinsa Corporation (Batch no. 129; East Windsor, NJ, USA). A 250 mM stock solution was prepared by dissolving GGMSC in sterile distilled water. For RNA sequencing experiments, CAPAN-2 and HPAF-II cells were treated with 500 µM GGMSC for either 8 or 20 h. Control cells were cultured under identical conditions without treatment. Each treatment group was performed in three independent biological replicates to ensure experimental consistency. After treatment, cells were harvested and stored at −80 °C for subsequent RNA extraction.

Cytotoxicity assays were performed to determine the half-maximal inhibitory concentration (IC_50_) of GGMSC. Cells were seeded in 96-well plates (BD Falcon, Durham, NC, USA) at a density of 400 cells/mm^2^ for both cell lines. After 24 h, cells were exposed to serial dilutions of GGMSC, with concentrations ranging up to 10 mM. Cell viability was measured at 24, 48, and 72 h using the CellTiter-Glo^®^ 2.0 Luminescent Cell Viability Assay (Promega, Madison, WI, USA), which quantifies intracellular ATP as an indicator of metabolic activity. Luminescence readings were collected using a CLARIOstar^®^ Luminometer (BMG Labtech, Ortenberg, Germany). Each assay was performed in triplicate, with five independent experiments conducted for each cell line. IC_50_ values were calculated using GraphPad Prism version 10.1.2 (GraphPad Software Inc., San Diego, CA, USA).

### 2.3. RNA Isolation and mRNA Library Construction

Total RNA was extracted from CAPAN-2, and HPAF-II cell lines using the Maxwell^®^ RSC simplyRNA Cells Kit (Promega, Madison, WI, USA), which employs paramagnetic bead-based purification. RNA concentration and purity were evaluated using a NanoDrop ND-1000 Spectrophotometer (Thermo Fisher Scientific, Waltham, MA, USA). cDNA libraries were prepared according to the SMART™-seq2 protocol, which utilizes a template-switching oligonucleotide (TSO_II) for full-length cDNA synthesis [15]. In brief, 200 ng of total RNA was reverse transcribed using 1 mM dNTP mix and 1 µM VN30 Poly-T oligonucleotide tagged at the 5’ end with a specific adaptor sequence (Table 1). The reaction mixture was incubated at 72 °C for 3 min and then cooled to 4 °C. Next, a second reaction mix was prepared containing 1× Template Switching RT Buffer, 3.75 µM Template Switching Oligo II (Integrated DNA Technologies, Leuven, Belgium), and 1× Template Switching RT Enzyme Mix (New England Biolabs, Ipswich, MA, USA). Both mixtures were combined and incubated at 42 °C for 90 min, followed by inactivation at 85 °C for 5 min. The resulting double-stranded cDNA was subjected to limited amplification using PCR_Oligo_II primers (Appendix A) and 1× Platinum SuperFi™ PCR Master Mix (Thermo Fisher Scientific, Waltham, MA, USA) in a 15 µL reaction. Amplification conditions are provided in Appendix A. Purification of amplified cDNA was performed using SPRIselect™ Bead-Based Reagent (Beckman Coulter Life Sciences, Indianapolis, IN, USA), and cDNA concentration was measured with a Qubit™ 4.0 Fluorometer (Thermo Fisher Scientific, Rockford, IL, USA).

### 2.4. Illumina Library Preparation and Sequencing

RNA-seq libraries were prepared using 50 ng of purified cDNA with the Illumina DNA Prep Kit (Illumina, San Diego, CA, USA) according to the manufacturer’s instructions. The protocol included tagmentation, post-tagmentation cleanup, PCR amplification, and pooling of indexed libraries. Library concentrations were determined using the Qubit™ 4.0 Fluorometer (Thermo Fisher Scientific, Rockford, IL, USA), and fragment sizes along with molarity were assessed using a Bioanalyzer™ 2100 equipped with the High Sensitivity DNA Kit (Agilent Technologies, Santa Clara, CA, USA). Final libraries were then normalized to 4 nM. Library denaturation and dilution were performed following the Illumina NextSeq System Denature and Dilute Libraries Guide (Illumina, San Diego, CA, USA). Sequencing was conducted on the Illumina NextSeq™ 500 platform, utilizing mid-output flow cells, generating approximately 130 million clusters per run in line with the manufacturer’s standard workflow.

### 2.5. Bioinformatics and Data Analysis

Transcriptome data were processed and analyzed using the Chipster virtual platform provided by the IT Center for Science (CSC), Finland [16]. Adapter sequences were trimmed, and sequencing read quality was assessed with FastQC. High-quality paired-end reads (15 to 20 million per sample) were aligned to the human reference genome (GRCh38.95) using the STAR aligner. Aligned reads (BAM files) were subsequently quantified at the gene level using HTSeq. Differential gene expression analysis was performed with the DESeq2 package from Bioconductor. Normalized count matrices from treated and control groups were combined for the calculation of log_2_ fold changes. Genes with an adjusted *p*-value (padj) ≤ 0.01 and absolute log_2_ fold change ≥ 1 were considered significantly differentially expressed.

Hierarchical clustering, heatmaps, and dendrograms were generated from raw gene counts normalized via DESeq2. Overrepresentation analysis of differentially expressed genes was performed using ConsensusPathDB, integrating pathways from over 30 publicly available databases. Venn diagrams illustrating overlaps in differentially expressed genes between conditions were created using Venny 2.1. Available online: https://bioinfogp.cnb.csic.es/tools/venny/ (accessed on 26 June 2025). For pathway enrichment and functional annotation, significantly differentially expressed genes (padj < 0.01) were analyzed using Enrichr. Available online: https://maayanlab.cloud/Enrichr/ (accessed on 30 June 2025) to identify enriched terms from Gene Ontology (Biological Process, Cellular Component, and Molecular Function), KEGG, Reactome, and FerrDb databases [17,18,19,20]. Additional pathway resources, including Pathway Commons and STRING, were also used for functional clustering and interaction analysis [21]. Transcription factor enrichment and regulatory network analyses were conducted using ChEA3, JASPAR, EPD-TALDO1, and ChIP-Atlas databases. These tools enabled the identification of potential upstream transcriptional regulators and binding motifs associated with the differentially expressed gene sets.

### 2.6. Statistical Analysis

Statistical analyses were conducted using GraphPad Prism version 10.1.2 (GraphPad Software, San Diego, CA, USA) where appropriate. Differential gene expressions from RNA-seq data was evaluated using the DESeq2 package within the Chipster platform, with statistical significance set at an adjusted *p*-value (padj) < 0.01. Cytotoxicity (IC_50_) values were calculated from dose-response curves, and results are presented as mean ± SD from at least five biologically independent experiments, each performed in triplicate.

## 3. Results

### 3.1. Global Transcriptomic Profiling Reveals Distinct Gene Expression Changes upon GGMSC Treatment in PDAC Cell Lines

To investigate the global transcriptional response to GGMSC, we performed RNA sequencing on CAPAN-2 (GSE309447), and HPAF-II (GSE309643) pancreatic cancer cell lines treated with 500 µM GGMSC for 20 h. Principal component analysis (PCA) demonstrated clear separation between treated and control samples in both cell lines, indicating robust transcriptional shifts following GGMSC exposure (Figure 1A,B). In CAPAN-2, the first principal component (PC1) accounted for 89% of the variance (Figure 1A), while in HPAF-II, PC1 explained 82% of the variance (Figure 1B). PCA plots for the 8 h treatment also showed distinct clustering between conditions, albeit with lower variance explained, and are provided in Appendix A. Differential gene expression analysis revealed a substantial number of significantly altered genes (adjusted *p*-value < 0.01) (Appendix A). In CAPAN-2 cells, volcano plots highlighted upregulation of genes including *GDF15* (ENSG00000130513), *HERPUD1* (ENSG00000051108), *HSPA5* (ENSG00000044574), *NNMT* (ENSG00000166741), and *CORO1A* (ENSG00000102879), while *PHF19* (ENSG00000119403), *PMF1* (ENSG00000160783), and *HMGN2* (ENSG00000198830) were among the most downregulated transcripts (Figure 1C). In HPAF-II cells, GGMSC treatment led to strong upregulation of *HOXA3* (ENSG00000105997), *RASSF3* (ENSG00000153179), *HNRNPH1* (ENSG00000169045), *EGR1* (ENSG00000120738), and *BROX* (ENSG00000162819), while *IDH2* (ENSG00000182054), *RPUSD3* (ENSG00000156990), *VGLL4* (ENSG00000144560), and *MDK* (ENSG00000110492) were markedly downregulated (Figure 1D). Quantitative comparison of differentially expressed genes at both 8 h and 20 h timepoints revealed a time-dependent increase in transcriptional alterations in both cell lines (Figure 1E,F). Cells exhibited a greater number of DEGs at 20 h compared to 8 h (Figure 1E), a pattern similarly observed in HPAF-II (Figure 1F).

### 3.2. Common and Cell Line-Specific Gene Expression Profiles Induced by GGMSC

To assess shared versus unique responses to GGMSC, we identified differentially expressed genes (DEGs) at both 8 h and 20 h timepoints using a stringent cutoff of adjusted *p*-value < 0.01. A summary of DEG counts for CAPAN-2 and HPAF-II is presented in Table 1. We then focused on representative genes consistently altered in both cell lines after 20 h of treatment. Bar plots of selected transcripts showed clear cell line–specific expression patterns, with genes like *ASF1A* (ENSG00000111875), *LRPAP1* (ENSG00000163956), and *S100A16* (ENSG00000188643) upregulated in both cell lines, and others, including *IDH2*, *CFD* (ENSG00000197766), *GALK1* (ENSG00000108479), and *TMED1* (ENSG00000099203) showing distinct regulation (Appendix A). We identified 392 commonly regulated genes across both cell lines (14.7% overlap), including *ASF1A* (ENSG00000111875), *LRPAP1*, *ERRFI1* (ENSG00000116285), and *LENG8* (ENSG00000167615), suggesting these may represent core components of the GGMSC response (Figure 2A). Applying an additional log2 fold change cutoff (|log2FC| > 1) refined this list to high-confidence DEGs involved in mitochondrial metabolism, RNA processing, and chromatin remodelling, including *IDH2*, *RPUSD3*, *VGLL4*, *MDK*, and *RAMP1* (ENSG00000132329) (Figure 2B). A subset of twelve genes, including *ERRFI1*, *IDH2*, *ASF1A*, *GALK1*, *S100A16, TMED1, TUBA1B* (ENSG00000123416), and *CFD*, showed coordinated expression between the two cell lines, reflecting a conserved transcriptional module triggered by GGMSC (Figure 2C).

### 3.3. GGMSC Modulates Functional Pathways Associated with Oxidative Stress, Metabolic Rewiring, and Chromatin Regulation

To uncover functional mechanisms underlying the transcriptional changes, we performed pathway enrichment analysis using DEGs from CAPAN-2 and HPAF-II at 20 h. Significantly enriched pathways were grouped into stress responses, redox regulation, mitochondrial function, and chromatin dynamics (Table 2). CAPAN-2 cells showed strong activation of the unfolded protein response (UPR), ER stress, and oxidative stress signalling, with upregulation of KEAP1-NFE2L2 and nuclear NRF2 target pathways. These effects were accompanied by enhanced expression of mitochondrial metabolism and ferroptosis-associated genes. In contrast, HPAF-II exhibited milder stress induction and downregulation of ferroptosis and glutathione metabolism genes, reflecting partial resistance. GGMSC also repressed cell cycle and DNA replication programs in CAPAN-2, consistent with the action of its active metabolite β-methylselenopyruvate (MSP), known to inhibit histone deacetylases (HDACs). Downregulation of genes involved in chromosome condensation and histone acetylation supports epigenetic repression. CAPAN-2 cells further showed enrichment of autophagy and ferroptosis programs, while HPAF-II downregulated apoptotic regulators, indicating a survival-oriented phenotype (Table 2).

### 3.4. Transcription Factor Responses Mirror Stress Sensitivity and Resistance

We next profiled differentially expressed transcription factors (TFs) to understand cell line–specific regulatory shifts. CAPAN-2 upregulated stress-associated TFs, including *DDIT3 (CHOP)* (ENSG00000175197), *ATF4* (ENSG00000128272), *NFE2L2 (NRF2)* (ENSG00000116044), and *BACH1* (ENSG00000156273), while suppressing *NFKB1*, suggesting a shift toward a pro-death, oxidative stress-driven program (Table 3). In contrast, HPAF-II upregulated *HOXA3, HNF4A* (ENSG00000101076), *CBX3* (ENSG00000122565), and *CHUK* (ENSG00000213341), consistent with epithelial identity and sustained NF-κB signalling. Downregulation of *DDIT3* in HPAF-II further indicates attenuation of the ER stress-induced apoptotic pathway (Table 3). These contrasting TF signatures are presented in Appendix A and support a model where CAPAN-2 activates transcriptional repression and death, while HPAF-II maintains survival pathways.

### 3.5. GGMSC Induces Metabolic Stress and Ferroptosis in CAPAN-2 Cells

We categorized DEGs by pathway to explore GGMSC’s effects on metabolism, stress, and regulated cell death. HPAF-II showed strong downregulation of metabolic genes across glycolysis, PPP, TCA cycle, and OXPHOS, consistent with suppressed energy metabolism (Figure 2D). CAPAN-2 retained metabolic gene expression, suggesting mitochondrial engagement (Figure 2D). Stress-related transcripts including *DDIT3*, *HIF1A* (ENSG00000100644), and *SLC2A1* (ENSG00000117394) were highly enriched in CAPAN-2 (Figure 3A). Appendix A shows additional oxidative stress genes (*TXNRD1,* (ENSG00000198431), *GLRX,* (ENSG00000173221), *HMOX1,* (ENSG00000100292)) selectively upregulated in CAPAN-2. CAPAN-2 also upregulated ferroptosis inducers (*ACSL4,* (ENSG00000068366), *CHAC1,* (ENSG00000128965), *ATF4, HMOX1*), while HPAF-II did not (Figure 3B). A broader analysis of cell death and survival genes revealed repression of ferroptosis inhibitors (*IDH2, ACSL3* (ENSG00000123983), *HSPA5*) and upregulation of apoptosis and pro-death signals in CAPAN-2 (Figure 3C). HPAF-II instead upregulated survival-promoting transcripts. Among these, *ACSL4* and *CHAC1* represent core ferroptosis effectors, directly contributing to lipid peroxidation and glutathione depletion, respectively, and are therefore most likely linked to execution of ferroptotic cell death. *ATF4* and *HMOX1*, while strongly upregulated, are considered part of broader integrated stress and antioxidant responses and may act upstream or in parallel to ferroptosis rather than serving as direct death effectors. Similarly, downregulation of *IDH2* and *ACSL3* removes metabolic buffers against lipid ROS and ferroptosis, whereas repression of *HSPA5* reflects ER stress engagement with potential crosstalk to ferroptotic sensitivity. Taken together, these changes suggest that CAPAN-2 engages both direct ferroptosis execution programs and secondary stress pathways that converge on enhanced vulnerability to cell death, while HPAF-II primarily sustains protective transcriptional states. GGMSC also influenced lipid, iron, and amino acid metabolism (Appendix A), with CAPAN-2 again showing stronger transcriptional activation. These results highlight broader metabolic vulnerability in CAPAN-2.

### 3.6. GGMSC Suppresses Epigenetic and Antioxidant Programs While Enhancing Stress Signalling

To assess how GGMSC affects chromatin and transcriptional regulation, we evaluated genes involved in epigenetic remodelling, cell cycle control, and NRF2/ATF4 signalling (Figure 4). CAPAN-2 cells showed downregulation of chromatin remodelers (*SMARCC2* (ENSG00000139613), *EZH2* (ENSG00000106462), *DNMT1* (ENSG00000130816)), histone modifiers (*ING5* (ENSG00000168395), *HMGN2*, *SETD7* (ENSG00000145391), *KAT6B* (ENSG00000156650)), and nucleosome assembly genes (*ATRX* (ENSG00000085224), *H2AZ1* (ENSG00000164032), *ANP32B* (ENSG00000136938)) (Figure 4A). Cell cycle regulators, including cyclins (*CCNA2* (ENSG00000145386), *CCNB2* (ENSG00000134057), *CCND1* (ENSG00000110092)), and *PSMC*s (*PSMC1* to *PSMC5*), and DNA replication machinery (*MCM3* (ENSG00000112118), *MCM4* (ENSG00000104738), *MCM6* (ENSG00000076003), *MCM10* (ENSG00000065328), *PLK2* (ENSG00000145632)), were suppressed in CAPAN-2 (Figure 4B), consistent with growth arrest.

CAPAN-2 also strongly upregulated NRF2 and ATF4 target genes involved in redox balance, ER stress, and amino acid metabolism (*HMOX1*, *GCLC* (ENSG00000001084), *TXNRD1*, *SLC7A11* (ENSG00000151012), *CHAC1*, *SLC1A5* (ENSG00000105281), *DDIT3*, *HERPUD1*, and *PSAT1* (ENSG00000135069)) (Figure 4C,D). These genes are central to the unfolded protein response, amino acid metabolism, and ferroptosis, supporting a transcriptional program shaped by ER stress and metabolic adaptation. In contrast, HPAF-II showed minimal induction of these programs. Cytotoxicity assays confirmed that CAPAN-2 was more sensitive to GGMSC than HPAF-II, consistent with its transcriptional profile of stress and death activation (Appendix A). These findings demonstrate that GGMSC induces profound metabolic, transcriptional, and epigenetic stress in CAPAN-2, while HPAF-II maintains resistance through retained antioxidant and proliferative signalling.

**Figure 3 antioxidants-14-01163-f003:**
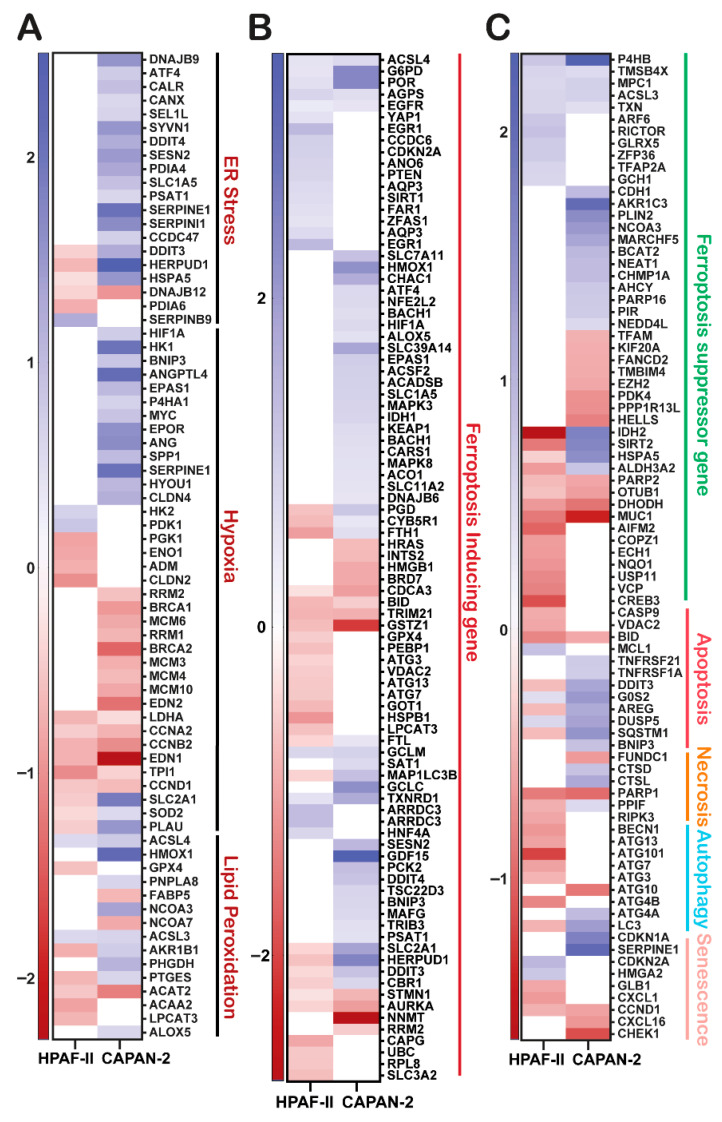
GGMSC reprograms cell death pathways in CAPAN-2 and HPAF-II cells. Heatmaps of differentially expressed genes (padj < 0.01), categorized by functional pathways. (**A**) Stress response genes linked to ER stress, hypoxia, and lipid peroxidation are highly upregulated in CAPAN-2. (**B**) Ferroptosis-inducing genes are selectively induced in CAPAN-2. (**C**) Genes related to ferroptosis suppression, apoptosis, autophagy, and senescence reveal a death-prone profile in CAPAN-2 and a survival-oriented transcriptional state in HPAF-II.

**Figure 4 antioxidants-14-01163-f004:**
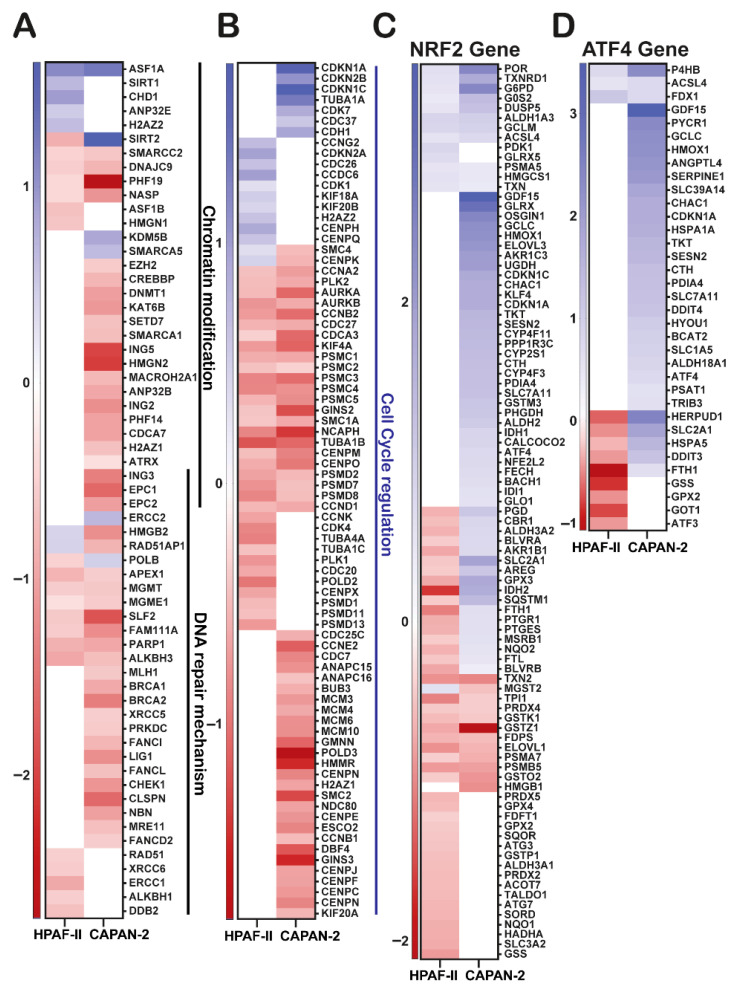
GGMSC disrupts chromatin remodeling, cell cycle, and stress-responsive transcriptional programs. (**A**) Epigenetic regulatory genes, including chromatin remodelling, histone modifiers, and DNA repair mechanism, are downregulated in CAPAN-2. (**B**) Expression of cell cycle and DNA replication genes are suppressed in CAPAN-2. (**C**) NRF2 target genes are upregulated in CAPAN-2 but not in HPAF-II. (**D**) ATF4-regulated genes follow a similar pattern, reflecting a CAPAN-2–specific stress response.

## 4. Discussion

Pancreatic ductal adenocarcinoma (PDAC) remains one of the most lethal malignancies, with a 5-year survival rate of less than 10%, posing a major clinical challenge and highlighting the urgent need for new therapeutic strategies [1,3]. The aggressive nature of PDAC, combined with its dense stromal environment and metabolic plasticity, contributes to poor drug penetration and therapeutic resistance [4,22]. Given these barriers, there is growing interest in agents targeting metabolic vulnerabilities and redox imbalances unique to PDAC.

Selenium-based compounds, including seleno-L-methylselenocysteine (MSC) and its structural analogues, have emerged as promising anticancer agents due to their ability to modulate redox balance and epigenetic regulation. MSC has been shown to induce apoptosis, inhibit histone deacetylases (HDACs), and sensitize cancer cells to ferroptosis and ER stress [7,23,24]. GGMSC, a glutamyl-conjugated derivative of MSC, retains similar biological activity but requires higher doses for activation, suggesting differential metabolic processing [10]. Despite limited literature specifically on GGMSC, its structural similarity to MSC and preliminary reports of redox activity and growth inhibition in various cancers make it a strong candidate for further investigation [25]. Supporting this, GGMSC has demonstrated anticancer efficacy in rat models, where it produced gene expression changes like MSC and inhibited tumour progression similar to MSC [12,13].

In this study, we investigated the transcriptomic response to high-dose GGMSC (500 µM) in two PDAC cell lines, CAPAN-2 and HPAF-II, and uncovered distinct mechanisms underlying their differential sensitivity. Our results demonstrate that GGMSC induces widespread gene expression changes in both cell lines, with CAPAN-2 exhibiting a robust stress response and cell death signature, while HPAF-II maintained metabolic and proliferative programs, indicative of resistance.

Oxidative stress, ferroptosis, and ER stress emerged as key themes in CAPAN-2 cells following GGMSC treatment. Ferroptosis-related genes were selected based on FerrDb and Enrichr pathway enrichment. Notably, genes regulated by the ATF4–CHOP (*DDIT3*) and NRF2 (*NFE2L2*) pathways were strongly upregulated, including *HMOX1*, *GCLC*, *TXNRD1*, and *CHAC1*. This dual activation of ATF4 and NRF2 suggests a coordinated response to redox and proteotoxic stress, consistent with studies demonstrating their synergistic role in promoting ferroptosis and apoptotic responses [9,26,27]. In addition, CAPAN-2 cells showed upregulation of ferroptosis inducers (e.g., *ACSL4*, *ALOX5*, *CHAC1*) and repression of ferroptosis suppressors, suggesting engagement of ferroptotic cell death pathways [28,29,30]. Mechanistically, the gene clusters induced by GGMSC in CAPAN-2 align with known regulators of ferroptosis, ER stress, and metabolic reprogramming. The upregulation of ATF4-DDIT3-CHAC1 and NRF2-HMOX1-SLC7A11 modules mirrors ferroptosis execution pathways, while suppression of TCA cycle and oxidative phosphorylation components suggests mitochondrial dysfunction. These patterns distinguish GGMSC from other selenium compounds that mainly drive apoptosis or cell cycle arrest, positioning GGMSC as a dual modulator of ferroptosis and metabolic shutdown [31]. In addition, GGMSC-induced metabolic reprogramming in CAPAN-2 was marked by downregulation of glycolytic enzymes, suppression of TCA cycle components, and reduced oxidative phosphorylation gene expression, suggesting energetic and biosynthetic constraints that heighten redox vulnerability.

In contrast, HPAF-II cells exhibited limited induction of redox stress response genes and maintained expression of core metabolic and cell cycle regulators. Upregulation of transcription factors, including *HOXA3*, *HNF4A*, and *CBX3*, along with NF-κB pathway component *CHUK*, suggests a more epithelial and proliferative phenotype. Downregulation of *DDIT3* (CHOP) and limited NRF2/ATF4 activation in HPAF-II may reflect protective transcriptional programs that buffer against oxidative and proteotoxic stress. Although direct evidence in PDAC is limited, low *CHOP* expression has been associated with poor clinical outcomes, implying resistance to ER stress-induced apoptosis and restrained NRF2 activity can foster adaptive stress responses without initiating cell death [32,33]. These mechanisms may partially explain HPAF-II’s reduced sensitivity to GGMSC.

Many of the transcriptional signatures observed with GGMSC resemble those previously reported for MSC, including NRF2 and ATF4 activation, HDAC inhibition, and ferroptosis induction [34]. In our prior study, MSC at lower concentrations (250 µM) triggered similar gene expression patterns in CAPAN-2 and PANC1 cells, implicating hKYAT1-mediated bioactivation as a key step [34]. GGMSC appears to require higher concentrations and longer treatment durations for similar effects, likely due to lower enzymatic cleavage efficiency, suggesting involvement of alternative enzymes in its activation.

We hypothesize that GGMSC functions as a prodrug, cleaved by γ-glutamyl transpeptidase (GGT) to release MSC, which is then metabolized to methylselenol (CH_3_;SeH) and β-methylselenopyruvate (MSP) by hKYAT1 [10]. Although this enzymatic cascade has not been experimentally validated for GGMSC in our study, but the hypothesis is supported by analogies to other γ-glutamyl selenium compounds. GGTs are well-known to catalyze the hydrolysis of γ-glutamyl bonds in xenobiotics and endogenous compounds (including glutathione and GPNA), releasing cysteinylglycine derivatives further processed by dipeptidases to yield active metabolites [35,36,37]. GGT-mediated cleavage thus remains a plausible activation route for GGMSC. It is possible that differential GGT activity between CAPAN-2 and HPAF-II could contribute to their distinct cytotoxic responses. We propose that GGMSC may exhibit selective toxicity toward tumours due to elevated GGT expression in many cancers, with lower GGT expression in normal tissues potentially sparing non-tumour cells [38,39]. GGT is frequently overexpressed in cancers and is linked to tumour progression, chemoresistance, and poor prognosis [38,39,40,41]. This supports the hypothesis that GGMSC activation may be enhanced in tumours with elevated GGT activity, potentially explaining the higher sensitivity of CAPAN-2.

The therapeutic implications of these findings are significant. Key proliferation markers, including *PCNA*, *MKI67*, *TOP2A*, *CHEK1*, and *UBE2C*, were markedly downregulated in CAPAN-2 following GGMSC treatment [14,42]. This highlights a robust cell cycle arrest, reinforcing GGMSC’s potential to suppress proliferation in aggressive PDAC. CAPAN-2’s vulnerability raises the possibility that biomarkers like high ATF4/NRF2 expression or ferroptosis sensitivity could be explored in further work for patient stratification. In contrast, HPAF-II’s resistance suggests that further studies could evaluate combinations of GGMSC with NF-κB inhibitors, ferroptosis enhancers, or chromatin modulators. The broad transcriptional reprogramming induced by GGMSC including suppression of cell cycle genes, epigenetic regulators, and mitochondrial metabolism supports its utility as a multi-targeted anticancer agent.

In addition to transcriptomic profiling, metabolomic approaches have provided valuable insights into cancer-associated metabolic deregulation. Comprehensive plasma metabolomic studies, by Soldevilla et al. in neuroendocrine tumors, have demonstrated the diagnostic and biological relevance of metabolic reprogramming in oncology [43]. Integrating metabolomic data with transcriptomic signatures in future work may further clarify the metabolic vulnerabilities exploited by GGMSC and strengthen translational relevance.

This study has several limitations that should be acknowledged. First, we investigated only two PDAC cell lines, which may not fully capture the heterogeneity of pancreatic tumors. Second, the transcriptomic profiling was performed at a relatively high GGMSC concentration (500 µM), reflecting the need for in vitro activation of this γ-glutamyl prodrug but limiting direct physiological extrapolation. Third, while our RNA-seq analyses revealed robust and cell line-specific transcriptional reprogramming, we did not perform protein-level or functional validation (e.g., ferroptosis assays, or GGT activity measurements). Future work in larger cell line panels, patient-derived models, and with complementary biochemical assays will be necessary to strengthen the mechanistic interpretation and translational relevance of these findings.

From clinical perspective, international guidelines (NCCN, ESMO, ASCO) recommend stage-specific, multidisciplinary management of PDAC, with systemic therapies such as modified FOLFIRINOX or gemcitabine/nab-paclitaxel as standard-of-care regimens, and more recently NALIRIFOX as an additional first-line option. Molecular testing for BRCA/PALB2 mutations, MSI-H/dMMR status, and NTRK fusions is recommended to guide precision therapy, while early integration of supportive care is strongly endorsed. Within this established therapeutic landscape, our study highlights GGMSC as an experimental redox-modulating strategy that may complement, rather than replace, existing treatment approaches.

In conclusion, our transcriptomic profiling reveals that GGMSC exerts potent cytotoxic effects in PDAC cells through coordinated activation of redox and stress-responsive pathways, particularly in CAPAN-2. The differential responses highlight the importance of understanding cellular context for therapeutic optimization and provide a rationale for exploring GGMSC in combination therapies for PDAC.

## Figures and Tables

**Figure 1 antioxidants-14-01163-f001:**
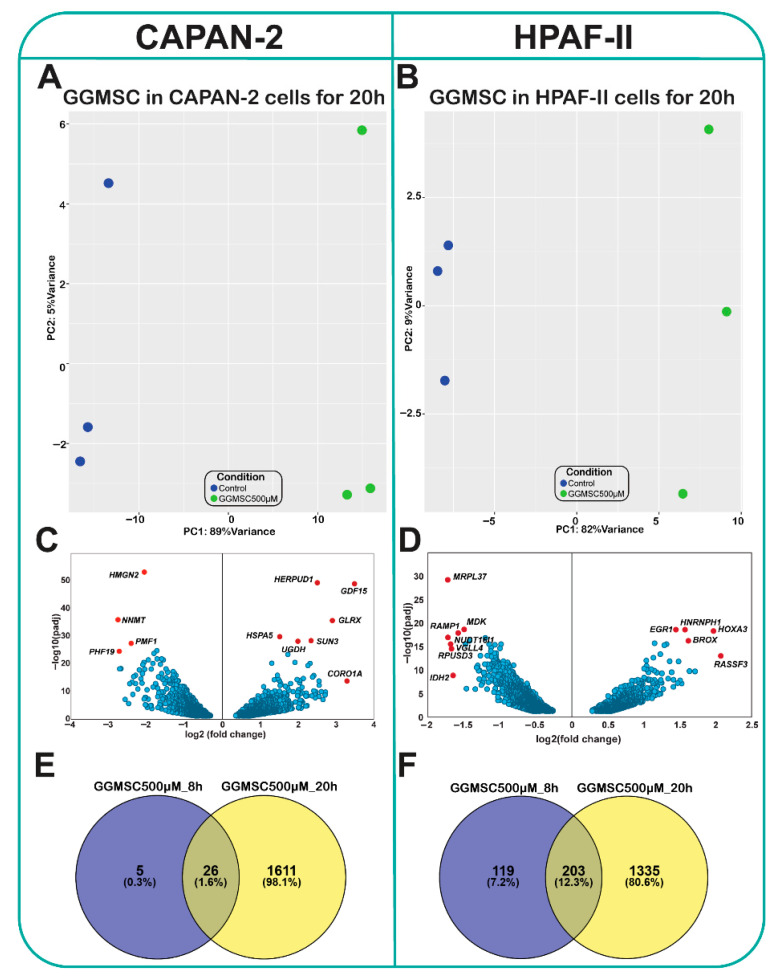
GGMSC induces distinct transcriptional responses in CAPAN-2 and HPAF-II pancreatic cancer cells. (**A**,**B**) Principal component analysis (PCA) of RNA-seq data showing clear separation between control and GGMSC-treated (500 µM, 20 h) samples in CAPAN-2 (**A**) and HPAF-II (**B**) cells, indicating robust treatment-induced transcriptomic shifts. (**C**,**D**) Volcano plots illustrating differentially expressed genes (padj < 0.01) in CAPAN-2 (**C**) and HPAF-II (**D**) cells after GGMSC treatment. Blue dots represent significantly up- or downregulated genes. Selected genes of interest are labelled. (**E**,**F**) Bar plots showing the number of differentially expressed genes (padj < 0.01) in CAPAN-2 (**E**) and HPAF-II (**F**) cells following GGMSC treatment for 8 h and 20 h, indicating a time-dependent increase in transcriptional alterations.

**Figure 2 antioxidants-14-01163-f002:**
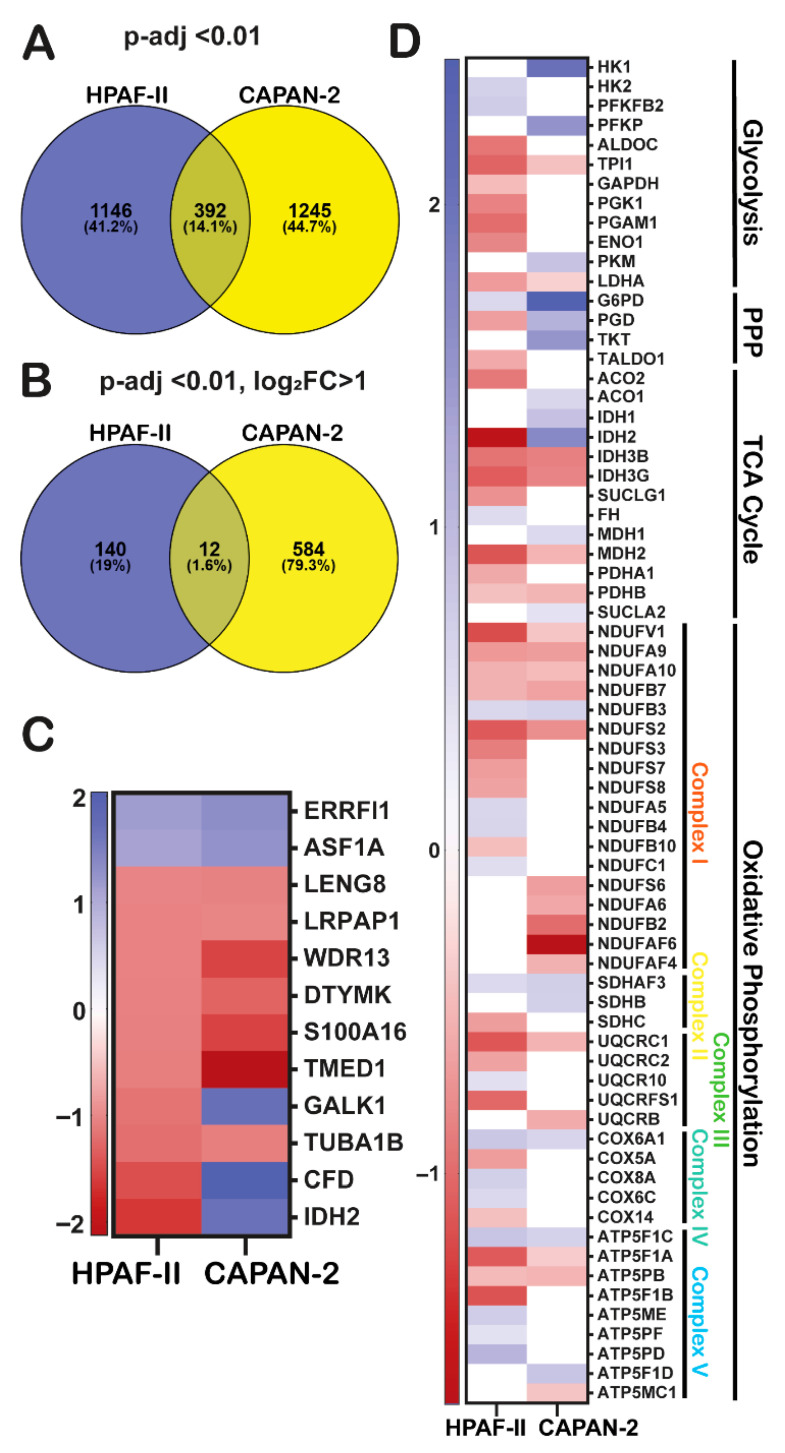
Overlapping and distinct transcriptional responses to GGMSC in CAPAN-2 and HPAF-II cells. (**A**) Venn diagram showing overlap of significantly regulated genes in both CAPAN-2 and HPAF-II, highlighting common targets (padj < 0.01). (**B**) Genes significantly upregulated in both cell lines at 20 h under a stringent threshold (padj < 0.01 and log_2_FC > 1). (**C**) Log_2_ fold changes of the 12 most significant and commonly regulated genes across both cell lines. GGMSC reprograms metabolic pathways in CAPAN-2 and HPAF-II cells. (**D**) Heatmaps of differentially expressed metabolic genes related to glycolysis, the pentose phosphate pathway (PPP), the tricarboxylic acid (TCA) cycle, and oxidative phosphorylation (OXPHOS), show broad suppression in HPAF-II and maintenance in CAPAN-2.

**Table 1 antioxidants-14-01163-t001:** Summary count of total DEGs per condition.

Cell Line	Condition	Total DEGs	Upregulation	Downregulation
CAPAN-2	GGMSC 500 µM-8 h	31	19	12
CAPAN-2	GGMSC 500 µM-20 h	1637	738	899
HPAF-II	GGMSC 500 µM-8 h	322	214	108
HPAF-II	GGMSC 500 µM-20 h	1538	557	981

**Table 2 antioxidants-14-01163-t002:** Functionally enriched pathways grouped by functional category in GGMSC-treated PDAC cell lines.

Biological Theme	Pathway	Source	Adjusted *p*-Value	Cell Line	Direction
Oxidative stress/NRF2	KEAP1-NFE2L2 pathway	Reactome	7.2 × 10^−8^	CAPAN-2	Up
	Nuclear events mediated by NFE2L2	Reactome	1.0 × 10^−7^	CAPAN-2	Up
	Glutathione metabolism	KEGG	0.0011	HPAF-II	Down
UPR/ER stress signaling	Protein processing in ER	KEGG	2.3 × 10^−5^	CAPAN-2	Up
	Response to ER stress (GO:0034976)	GO	3.9 × 10^−4^	CAPAN-2	Up
Mitochondrial/metabolic	Mitochondrial matrix (GO:0005759)	GO	3.9 × 10^−5^	CAPAN-2	Up
	Aerobic respiration and ETC	Reactome	3.3 × 10^−5^	HPAF-II	Up
	Glycolysis/gluconeogenesis	KEGG	1.8 × 10^−5^	HPAF-II	Down
Cell cycle/DNA repair	Cell cycle	Reactome	CAPAN-2: 1.2 × 10^−30^; HPAF-II: 6.4 × 10^−12^	CAPAN-2, HPAF-II	Down
	DNA replication	Reactome	1.5 × 10^−14^	CAPAN-2	Down
	Chromosome condensation	GO	1.3 × 10^−5^	CAPAN-2	Down
	Chromosome organization	GO	3.6 × 10^−10^	CAPAN-2	Down
	DNA repair	GO	1.2 × 10^−9^	CAPAN-2	Down
Cell death & survival	Ferroptosis	KEGG	CAPAN-2:0.00013; HPAF-II: 0.0015	CAPAN-2 (Up), HPAF-II (Down)	Mixed
	Autophagy	KEGG	0.00405	CAPAN-2	Up
	Regulation of apoptosis	Reactome	6.4 × 10^−11^	HPAF-II	Down

Significantly enriched pathways (adjusted *p*-value < 0.01) were identified in CAPAN-2 and HPAF-II cells treated with 500 µM GGMSC for 20 h. Pathways are organized into five major biological themes: (1) Oxidative stress and NRF2 signaling, (2) ER stress and proteostasis, (3) Mitochondrial and metabolic regulation, (4) Cell cycle and DNA repair, and (5) Cell death and survival. Adjusted *p*-values for each cell line are shown individually when available. Direction of regulation indicates whether genes contributing to the pathway were predominantly upregulated or downregulated. Notably, the observed downregulation of cell cycle- and chromatin-related pathways is consistent with GGMSC’s function as a methylselenol prodrug (MSP), potentially acting through HDAC inhibition and epigenetic modulation.

**Table 3 antioxidants-14-01163-t003:** Comparative analysis of transcription factor regulation in CAPAN-2 and HPAF-II cells following GGMSC treatment.

Transcription Factor	log_2_FC (CAPAN-2)	log_2_FC (HPAF-II)	Function	Interpretation
ATF4	+0.77 ↑	↔	Unfolded protein response	Activated only in CAPAN-2
NFE2L2	+0.77 ↑	↔	Oxidative stress (NRF2)	Activated in CAPAN-2
BACH1	+0.68 ↑	↔	Ferroptosis regulation	May balance NRF2; ferroptosis involvement
MYC	+0.88 ↑	↔	Proliferation/oncogene	Unexpected; could reflect transient response
NFKB1	−1.35 ↓	↔	NF-κB (survival)	Suppressed in CAPAN-2
DDIT3	+1.23 ↑	−0.44 ↓	ER stress/apoptosis	Pro-death activation in CAPAN-2; suppressed in HPAF-II
FOS	↔	−0.56 ↓	stress/proliferation	Suppressed in HPAF-II
HOXA3	↔	+1.97 ↑	Epithelial identity	Upregulated in HPAF-II
HNF4A	↔	+0.84 ↑	Metabolism/differentiation	Supports resistance and metabolic balance
CBX3	↔	+1.01 ↑	Chromatin regulation	May contribute to epigenetic resistance
CHUK	↔	+1.47 ↑	NF-κB signaling	Pro-survival signaling retained in HPAF-II

↑ indicates upregulation; ↓ indicates downregulation; ↔ indicates no significant change.

## Data Availability

The datasets generated and/or analyzed during the current study, including RNA-sequencing data are available from the corresponding author upon request.

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
