# Peer review of "Investigating the Molecular Impact of GGMSC on Redox and Metabolic Pathways in Pancreatic Cancer Cells"

_antioxidants, 2025, doi:10.3390/antiox14101163_

Round 1
Reviewer 1 Report
Introduction
The background on PDAC and selenium compounds is appropriate, but the section could benefit from a clearer summary of previous findings on GGMSC in cancers beyond PDAC.
The rationale for choosing only two cell lines (CAPAN-2 and HPAF-II) should be better justified, highlighting their biological differences.
Materials and Methods
The serum re-adaptation protocol is described in great detail. Please clarify whether this method has been validated in previous transcriptomic studies.
The chosen concentration of GGMSC (500 µM) seems relatively high. A stronger justification for this dose, especially regarding physiological relevance, would be helpful.
RNA-seq data availability should be improved. Instead of “available upon request,” the dataset should be deposited in a public repository (e.g., GEO, ArrayExpress).
Results
In section 3.5, on ferroptosis, it would be valuable to clarify which transcriptional changes are most directly linked to cell death versus secondary stress responses.
Discussion
The discussion provides good context and comparisons to MSC studies, but some parts are speculative. For example, the role of γ-glutamyl transpeptidase (GGT) in GGMSC activation is hypothesized but not experimentally tested here. This should be more clearly distinguished as a hypothesis.
A section on limitations would be useful, acknowledging issues such as: (i) use of only two PDAC cell lines, (ii) reliance on high compound concentrations, (iii) lack of protein-level or functional validation.
Conflict of Interest
The conflict of interest statement is transparent. However, given that a co-author is CEO of the company providing GGMSC, it would be reassuring to explicitly describe how independence in data analysis and interpretation was maintained.
Introduction
The background on PDAC and selenium compounds is appropriate, but the section could benefit from a clearer summary of previous findings on GGMSC in cancers beyond PDAC.
The rationale for choosing only two cell lines (CAPAN-2 and HPAF-II) should be better justified, highlighting their biological differences.
Materials and Methods
The serum re-adaptation protocol is described in great detail. Please clarify whether this method has been validated in previous transcriptomic studies.
The chosen concentration of GGMSC (500 µM) seems relatively high. A stronger justification for this dose, especially regarding physiological relevance, would be helpful.
RNA-seq data availability should be improved. Instead of “available upon request,” the dataset should be deposited in a public repository (e.g., GEO, ArrayExpress).
Results
In section 3.5, on ferroptosis, it would be valuable to clarify which transcriptional changes are most directly linked to cell death versus secondary stress responses.
Discussion
The discussion provides good context and comparisons to MSC studies, but some parts are speculative. For example, the role of γ-glutamyl transpeptidase (GGT) in GGMSC activation is hypothesized but not experimentally tested here. This should be more clearly distinguished as a hypothesis.
A section on limitations would be useful, acknowledging issues such as: (i) use of only two PDAC cell lines, (ii) reliance on high compound concentrations, (iii) lack of protein-level or functional validation.
Conflict of Interest
The conflict of interest statement is transparent. However, given that a co-author is CEO of the company providing GGMSC, it would be reassuring to explicitly describe how independence in data analysis and interpretation was maintained.
Author Response
Reviewer 1
We sincerely thank Reviewer 1 for their valuable comments and constructive suggestions. In response, we have clarified the background on GGMSC beyond PDAC, provided a stronger rationale for the chosen cell lines, explained the dose selection of 500 µM GGMSC, and revised the Discussion to clearly distinguish hypotheses (e.g., the role of GGT) from experimental findings. These changes have substantially improved the clarity and balance of the manuscript.
(1) Reviewer comment:
“The background on PDAC and selenium compounds is appropriate, but the section could benefit from a clearer summary of previous findings on GGMSC in cancers beyond PDAC.”
Response:
We thank the reviewer for this valuable suggestion. We have revised the Introduction to more clearly summarize the limited but relevant studies on GGMSC beyond PDAC. Specifically, we now describe the findings of Tung et al. (2015), who tested GGMSC in COLO205 colon adenocarcinoma cells up to 250 µM, the author did not observe any cytotoxic effects, in contrast to MSC, which triggered apoptosis via ER stress and cell death receptor pathways. We also highlight the in vivo study by Dong et al. (2001), which demonstrated that GGMSC, isolated from garlic, is bioavailable and exhibits elimination kinetics and tumor-suppressive activity comparable to MSC in rodent models. These additions clarify the current state of knowledge on GGMSC outside PDAC and underscore the rationale for our present study. The revised text is found in the introduction (line 66-74).
(2) Reviewer comment:
“The rationale for choosing only two cell lines (CAPAN-2 and HPAF-II) should be better justified, highlighting their biological differences.”
Response:
We have revised the Introduction to clarify our rationale for selecting these two PDAC models. Specifically, we now highlight that CAPAN-2 is a well-differentiated adenocarcinoma line derived from a primary pancreatic tumor and carries common PDAC driver mutations including KRAS, TP53, and SMAD4. In contrast, HPAF-II is a well-differentiated line derived from metastatic ascites, also harboring KRAS and TP53 mutations, and is characterized by strong epithelial morphology, with metastatic potential. These distinctions provided the rationale for comparing GGMSC responses in primary- versus metastasis-derived PDAC contexts. The revised text is now included in the introduction (line 77-85).
(3) Reviewer comment:
“The serum re-adaptation protocol is described in great detail. Please clarify whether this method has been validated in previous transcriptomic studies.”
Response:
We have clarified in the methods section that serum starvation and refeeding protocols have been used in prior transcriptomic studies to synchronize cancer cells and reduce variability due to cell cycle heterogeneity (Whitfield et al., 2002). This reference supports the validity of our approach. The revised text is found in the methods section (line:108-110).
(4) Reviewer comment:
“The chosen concentration of GGMSC (500 µM) seems relatively high. A stronger justification for this dose, especially regarding physiological relevance, would be helpful.”
Response:
We appreciate the reviewer’s concern regarding dose selection. We selected 500 µM GGMSC for transcriptomic profiling based on both literature evidence and our own pilot range-finding (cytotoxicity assays and morphology visualization). First, GGMSC is a γ-glutamyl prodrug that requires enzymatic cleavage (via γ-glutamyltransferase, GGT) to release MSC-like active metabolites. Such enzyme activity is often attenuated in standard cell culture, reducing apparent potency and necessitating higher nominal concentrations in vitro. Secondly, the only in vitro comparison we are aware of, GGMSC (reported as “GMSeC”) showed minimal cytotoxicity up to 250 µM in COLO205 cells, whereas MSC induced apoptosis, consistent with lower activation efficiency of the γ-glutamyl prodrug in culture. Together, these observations support the use of a higher nominal concentration (500 µM) to elicit measurable transcriptional responses in vitro, despite GGMSC showing in vivo activity at lower dietary exposures. The revised text is now included in the introduction (line 77-85).
(5) Reviewer comment:
RNA-seq data availability should be improved. Instead of “available upon request,” the dataset should be deposited in a public repository (e.g., GEO, ArrayExpress).
Response:
At this stage, the only pending item is the deposition of the RNA-seq dataset in the GEO repository. This process may require a little more time to complete, but we are working on it with the assistance of a co-author.
(6) Reviewer comment:
“In section 3.5, on ferroptosis, it would be valuable to clarify which transcriptional changes are most directly linked to cell death versus secondary stress responses.”
Response:
We have revised section 3.5 in result section to distinguish between direct ferroptosis effectors and secondary stress-responsive genes. Specifically, ACSL4 and CHAC1 are highlighted as core ferroptosis drivers with direct roles in lipid peroxidation and glutathione depletion, whereas ATF4 and HMOX1 are described as stress-responsive regulators that may act upstream or in parallel to ferroptosis. Similarly, repression of IDH2 and ACSL3 is interpreted as loss of metabolic defenses, while HSPA5 downregulation reflects ER stress involvement. This clarification helps separate transcriptional programs most directly linked to ferroptotic death from broader stress responses (Result: line 328-339).
(7) Reviewer comment:
“The discussion provides good context and comparisons to MSC studies, but some parts are speculative. For example, the role of γ-glutamyl transpeptidase (GGT) in GGMSC activation is hypothesized but not experimentally tested here. This should be more clearly distinguished as a hypothesis.”
Response:
We thank the reviewer for pointing this out. We have revised the discussion to clearly distinguish hypothesis-driven interpretations from experimental findings. In particular, the role of GGT in GGMSC activation is now explicitly stated as a hypothesis. Similarly, statements regarding differential GGT activity in CAPAN-2 versus HPAF-II, tumor-selective toxicity, potential biomarkers for patient stratification, and the suggestion of combination therapies have all been rephrased to emphasize that these are speculative proposals for future investigation rather than conclusions from the current study (Discussion: line 445-469).
(8) Reviewer comment:
“A section on limitations would be useful, acknowledging issues such as: (i) use of only two PDAC cell lines, (ii) reliance on high compound concentrations, (iii) lack of protein-level or functional validation.”
Response:
We have added a dedicated limitations paragraph at the end of the discussion. This section now explicitly acknowledges the restricted number of PDAC cell lines, the relatively high GGMSC concentration required in vitro, and the lack of protein-level or functional validation assays in the present study. We also indicate how future work could address these issues (Discussion: line 480-489).
(9) Reviewer comment:
“The conflict of interest statement is transparent. However, given that a co-author is CEO of the company providing GGMSC, it would be reassuring to explicitly describe how independence in data analysis and interpretation was maintained.”
Response:
We have expanded the Conflict of Interest statement to explicitly clarify that although S.M. (Sabinsa Corporation) provided the GGMSC compound, the company had no involvement in study design, data collection, analysis, interpretation, or the decision to publish. All experimental work and data analyses were conducted independently at Karolinska Institutet. The revised Conflict of Interest statement is now included in the manuscript (Conflicts of Interest: line 556-560).
Reviewer 2 Report
The quality of the manuscript is overall fine, but minor points must be addressed by the authors.
1) available guidelines for the management of PDAC must be added and discussed.
2) a graphical abstract should be added to improve clarity and give a direct message to the readers.
3) the limitations of the study and the comparison of the results achieved with the International standard should be discussed to put these data in the real world context.
4) metabolomics approach in oncology together with the metabolism deregulation must be discussed. The following reference should be cited Soldevilla B, López-López A, Lens-Pardo A, et al. Comprehensive Plasma Metabolomic Profile of Patients with Advanced Neuroendocrine Tumors (NETs). Diagnostic and Biological Relevance. Cancers (Basel). 2021 May 27;13(11):2634. doi: 10.3390/cancers13112634. PMID: 34072010; PMCID: PMC8197817.
5) few typos across the text should be corrected
See minor comments provided above
Author Response
Reviewer 2
We sincerely thank Reviewer 2 for their insightful feedback. In response, we have added a limitations and clinical context paragraph and included a discussion of metabolomics as a complementary approach, citing Soldevilla et al. (2021). These revisions further strengthen the manuscript and place our findings in a broader biomedical framework.
(1) Reviewer comment:
“Available guidelines for the management of PDAC must be added and discussed.”
Response:
We thank the reviewer for this suggestion. To provide appropriate clinical context, we have added a concise paragraph in the discussion noting that current international guidelines (NCCN, ESMO, ASCO) recommend stage-specific, multidisciplinary management of PDAC, with systemic therapies including modified FOLFIRINOX or gemcitabine/nab-paclitaxel as standard-of-care. We then emphasize that our study positions GGMSC as an experimental redox-modulating approach that may complement existing treatments, rather than replace them. (Discussion, line 490-498).
(2) Reviewer comment:
“A graphical abstract should be added to improve clarity and give a direct message to the readers.”
Response:
A graphical abstract was prepared and included with our original submission. To avoid any confusion, we have re-uploaded it as a separate file designated as “Graphical Abstract” in the revised submission.
(3) Reviewer comment:
“The limitations of the study and the comparison of the results achieved with the international standard should be discussed to put these data in the real-world context.”
Response:
We have added a new paragraph in the discussion that explicitly acknowledges the study’s limitations (restricted number of PDAC cell lines, the relatively high GGMSC concentration, and lack of protein-level or functional validation). In the same section, we also summarize international guideline-based standards for PDAC management (NCCN, ESMO, ASCO) to place our findings in clinical context. This addition clarifies that our results provide mechanistic insight into GGMSC as an experimental redox-modulating strategy that could complement existing therapeutic approaches. The new text is located in the discussion (Discussion, line 480-498).
(4) Reviewer comment:
“A metabolomics approach in oncology together with the metabolism deregulation must be discussed. The following reference should be cited: Soldevilla B, López-López A, Lens-Pardo A, et al. Comprehensive Plasma Metabolomic Profile of Patients with Advanced Neuroendocrine Tumors (NETs). Cancers (Basel). 2021 May 27;13(11):2634.”
Response:
We have added a short paragraph in the discussion noting that metabolomics represents a complementary approach to investigate metabolic deregulation in cancer. We now cite the work of Soldevilla et al. (Cancers 2021) as an example of how metabolomic profiling can reveal clinically relevant metabolic reprogramming. We emphasize that integration of metabolomic and transcriptomic approaches in future studies may help clarify the metabolic vulnerabilities targeted by GGMSC. (Discussion, line 473-479).
(5) Reviewer comment:
“Few typos across the text should be corrected.”
Response:
We thank the reviewer for noting this. We have carefully re-read the entire manuscript and corrected minor typographical and formatting errors throughout the text.